# SpArSe: Sparse Architecture Search for CNNs on Resource-Constrained Microcontrollers

**Igor Fedorov**
Arm ML Research
igor.fedorov@arm.com

**Ryan P. Adams**
Princeton University
rpa@princeton.edu

**Matthew Mattina**
Arm ML Research
matthew.mattina@arm.com

**Paul N. Whatmough**
Arm ML Research
paul.whatmough@arm.com

## Abstract

The vast majority of processors in the world are actually microcontroller units (MCUs), which find widespread use performing simple control tasks in applications ranging from automobiles to medical devices and office equipment. The Internet of Things (IoT) promises to inject machine learning into many of these every-day objects via tiny, cheap MCUs. However, these resource-impoverished hardware platforms severely limit the complexity of machine learning models that can be deployed. For example, although convolutional neural networks (CNNs) achieve state-of-the-art results on many visual recognition tasks, CNN inference on MCUs is challenging due to severe memory limitations. To circumvent the memory challenge associated with CNNs, various alternatives have been proposed that do fit within the memory budget of an MCU, albeit at the cost of prediction accuracy. This paper challenges the idea that CNNs are not suitable for deployment on MCUs. We demonstrate that it is possible to automatically design CNNs which generalize well, while also being small enough to fit onto memory-limited MCUs. Our Sparse Architecture Search method combines neural architecture search with pruning in a single, unified approach, which learns superior models on four popular IoT datasets. The CNNs we find are more accurate and up to $7.4\times$ smaller than previous approaches, while meeting the strict MCU working memory constraint.

## 1 Introduction

The microcontroller unit (MCU) is a truly ubiquitous computer. MCUs are self-contained single-chip processors which are small ($\sim 1\text{cm}^2$), cheap ($\sim \$1$), and power efficient ($\sim 1\text{mW}$). Applications are extremely broad, but often include seemingly banal tasks such as simple control and sequencing operations for everyday devices like washing machines, microwave ovens, and telephones. The key advantage of MCUs over application specific integrated circuits is that they are programmed with software and can be readily updated to fix bugs, change functionality, or add new features. The short time to market and flexibility of software has led to the staggering popularity of MCUs. In the developed world, a typical home is likely to have around four general-purpose microprocessors. In contrast, the number of MCUs is around three dozen [46]. A typical mid-range car may have about 30 MCUs. Public market estimates suggest that around 50 billion MCU chips will ship in 2019 [1], which far eclipses other chips like graphics processing units (GPUs), whose shipments totalled roughly 100 million units in 2018 [2].

MCUs can be highly resource constrained; Table 1 compares MCUs with bigger processors. The broad proliferation of MCUs relative to desktop GPUs and CPUs stems from the fact that they are

Table 1: Processors for ML inference: estimated characteristics to indicate the relative capabilities.

| Processor | Usecase | Compute | Memory | Power | Cost |
|---|---|---|---|---|---|
| Nvidia 1080Ti GPU | Desktop | 10 TFLOPs/Sec | 11 GB | 250 W | $700 |
| Intel i9-9900K CPU | Desktop | 500 GFLOPs/Sec | 256 GB | 95 W | $499 |
| Google Pixel 1 (Arm CPU) | Mobile | 50 GOPs/Sec | 4 GB | ~5 W | – |
| Raspberry Pi (Arm CPU) | Hobbyist | 50 GOPs/Sec | 1 GB | 1.5 W | – |
| Micro Bit (Arm MCU) | IoT | 16 MOPs/Sec | 16 KB | ~1 mW | $1.75 |
| Arduino Uno (Microchip MCU) | IoT | 4 MOPs/Sec | 2 KB | ~1 mW | $1.14 |

orders of magnitude cheaper ($\sim 600\times$) and less power hungry ($\sim 250,000\times$). In recent years, MCUs have been used to inject intelligence and connectivity into everything from industrial monitoring sensors to consumer devices, a trend commonly referred to as the Internet of Things (IoT) [10, 22, 43]. Deploying machine learning (ML) models on MCUs is a critical part of many IoT applications, enabling local autonomous intelligence rather than relying on expensive and insecure communication with the cloud [9]. In the context of supervised visual tasks, state-of-the-art (SOTA) ML models typically take the form of convolutional neural networks (CNNs) [35]. While tools for deploying CNNs on MCUs have started to appear [7, 6, 4], the CNNs themselves remain far too large for the memory-limited MCUs commonly used in IoT devices. In the remainder of this work, we use MCU to refer specifically to IoT-sized MCUs, like the Micro Bit. In contrast to this work, the majority of preceding research on compute/memory efficient CNN inference has targeted CPUs and GPUs [26, 11, 61, 62, 45, 54, 49].

To illustrate the challenge of deploying CNNs on MCUs, consider the seemingly simple task of deploying the well-known LeNet CNN on an Arduino Uno to perform MNIST character recognition [38]. Assuming the weights can be quantized to 8-bit integers, 420 KB of memory is required to store the model parameters, which exceeds the Uno's 32 KB of read-only (flash) memory. An additional 391 (resp. 12) KB of random access memory (RAM) is then required to store the intermediate feature maps produced by LeNet under memory model (5) (resp. (6)), which far exceeds the Uno's 2 KB RAM. The dispiriting implication is that it is not possible to perform LeNet inference on the Uno. This has led many to conclude that CNNs should be abandoned on constrained MCUs [36, 24]. Nevertheless, the sheer popularity of MCUs coupled with the dearth of techniques for leveraging CNNs on MCUs motivates our work, where we take a step towards bridging this gap.

Deployment of CNNs on MCUs is challenging along multiple dimensions, including power consumption and latency, but as the example above illustrates, it is the hard memory constraints that most directly prohibit the use of these networks. MCUs typically include two types of memory. The first is static RAM, which is relatively fast, but volatile and small in capacity. RAM is used to store intermediate data. The second is flash memory, which is non-volatile and larger than RAM; it is typically used to store the program binary and any constant data. Flash memory has very limited write endurance, and is therefore treated as read-only memory (ROM). The two MCU memory types introduce the following constraints on CNN model architecture:

**C1** : The maximum size of intermediate feature maps cannot exceed the RAM capacity.

**C2** : The model parameters must not exceed the ROM (flash memory) capacity.

To the best of our knowledge, there are currently no CNN architectures or training procedures that produce CNNs satisfying these memory constraints for MCUs with less than 2 KB RAM and deployed using standard toolchains [36, 24]. This is true even ignoring the memory required for the runtime (in RAM) and the program itself (in ROM). The severe memory constraints for inference on MCUs have pushed research away from CNNs and toward simpler classifiers based on decision trees and nearest neighbors [36, 24]. We demonstrate for the first time that it is possible to design CNNs that are at least as accurate as Kumar et al. [36], Gupta et al. [24] and at the same time satisfy **C1**-**C2**, even for devices with just 2 KB of RAM. We achieve this result by designing CNNs that are heavily specialized for deployment on MCUs using a method we call *Sparse Architecture Search* (SpArSe). The key insight is that combining neural architecture search (NAS) and network pruning allows us to balance generalization performance against tight memory constraints **C1**-**C2**. Critically, we enable SpArSe to search over pruning strategies in conjunction with conventional hyperparameters around morphology and training. Pruning enables SpArSe to quickly evaluate many sub-networks of a given

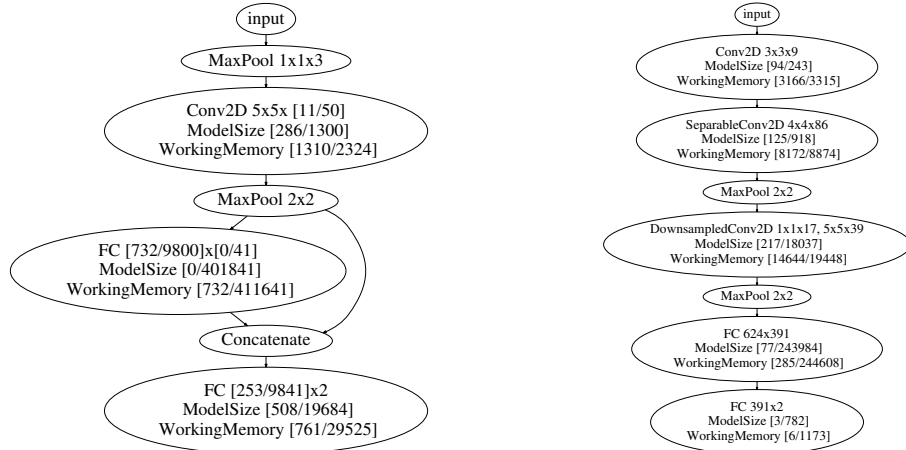

(a) Acc = 73.84%, MS = 1.31 KB, WM = 1.28 KB          (b) Acc=73.58%, MS = 0.61 KB, WM = 14.3 KB

Figure 1: Model architectures found with best test accuracy on CIFAR10-binary, while optimizing for (a) 2KB for both MODELSIZE (MS) and WORKINGMEMORY (WM), and (b) minimum MS. Each node in the graph is annotated with MS and WM using the model in (5), and the values in square brackets show the quantities before and after pruning, respectively. Optimizing for WM yields more than 11.2x WM reduction. Note that pruning has a considerable impact on the CNN.

network, thereby expanding the scope of the overall search. While previous NAS approaches have automated the discovery of performant models with reduced parameterizations, we are the first to simultaneously consider performance, parameter memory constraints, and inference-time working memory constraints.

We use SpArSe to uncover SOTA models on four datasets, in terms of accuracy and model size, outperforming both pruning of popular architectures and MCU-specific models [36, 24]. The multi-objective approach of SpArSe leads to new insights in the design of memory-constrained architectures. Fig. 1a shows an example of a discovered architecture which has high accuracy, small model size, and fits within 2KB RAM. By contrast, we find that optimizing networks solely to minimize the number of parameters (as is typically done in the NAS literature, e.g., [14]), is not sufficient to identify networks that minimize RAM usage. Fig. 1b illustrates one such example.

## 1.1  Related work

CNNs designed for resource constrained inference have been widely published in recent years [49, 30, 63], motivated by the goal of enabling inference on mobile phone platforms [60, 29]. Advances include depth-wise separable layers [50], deployment-centric pruning [62, 45], quantization [58, 21], and matrix decomposition techniques [55]. More recently, NAS has been leveraged to achieve even more efficient networks on mobile phone platforms [11, 52]. In a complimentary line of work, Gural and Murmann [25] propose memory-optimal direct convolutions (MODC). Unlike MODC, SpArSe yields CNNs that can be deployed with off-the-shelf tools and is shown to work on an array of IoT datasets.

Although mobile phones are more constrained than general-purpose CPUs and GPUs, they still have many orders of magnitude more memory capacity and compute performance than MCUs (Table 1). In contrast, little attention has been paid to running CNNs on MCUs, which represent the most numerous compute platform in the world. Kumar et al. [36] propose Bonsai, a pruned shallow decision tree with non-axis aligned decision boundaries. Gupta et al. [24] propose a compressed k-nearest neighbors (kNN) approach (ProtoNN), where model size is reduced by projecting data into a low-dimensional space, maintaining a subset of prototypes to classify against, and pruning parameters. We build upon Kumar et al. [36], Gupta et al. [24] by targeting the same MCUs, but using NAS to find CNNs which are at least as small and more accurate.

Algorithms for identifying performant CNN architectures have received significant attention recently [64, 14, 11, 40, 23, 15, 39]. The approaches closest to SpArSe are Stamoulis et al. [52], Elsken et al.

[14]. In Stamoulis et al. [52], the authors optimize the kernel size and number of feature maps of the MBConv layers in a MobileNetV2 backbone [49] by expressing each of the layer choices as a pruned version of a superkernel. In some ways, Stamoulis et al. [52] is less a NAS algorithm and more of a structured pruning approach, given that the only allowed architectures are reductions of MobileNetV2. SpArSe does not constrain architectures to be pruned versions of a baseline, which can be too restrictive of an assumption for ultra small CNNs. SpArSe is not based on an existing backbone, giving it greater flexibility to extend to different problems. Like Elsken et al. [14], SpArSe uses a form of weight sharing called network morphism [59] to search over architectures without training each one from scratch. SpArSe extends the concept of morphisms to expedite training and pruning CNNs. Because Elsken et al. [14] seek compact architectures by using the number of network edges as one of the objectives in the search, potential gains from weight sparsity are ignored, which can be significant (Section 3 [18, 19]). Moreover, since SpArSe optimizes both the architecture and weight sparsity, Elsken et al. [14] can be seen as a special case of SpArSe.

## 2 SpArSe framework: CNN design as multi-objective optimization

Our approach to designing a small but performant CNN is to specify a multi-objective optimization problem that balances the competing criteria. We denote a point in the design space as $\Omega = \{\alpha, \vartheta, \omega, \theta\}$, in which: $\alpha = \{V, E\}$ is a directed acyclic graph describing the network connectivity, where $V$ and $E$ denote the set of graph vertices and edges; $\omega$ denotes the network weights; $\vartheta$ represents the operations performed at each edge, i.e. convolution, pooling, etc.; and $\theta$ are hyperparameters governing the training process. The vertices $v_i, v_j \in V$ represent network neurons, which are connected to each other if $(v_i, v_j) \in E$ through an operation $\vartheta_{ij}$ parameterized by $\omega$. The competing objectives in the present work of targeting constrained MCUs are:

$$f_1(\Omega) = 1 - \text{VALIDATIONACCURACY}(\Omega) \qquad (1)$$

$$f_2(\Omega) = \text{MODELSIZE}(\omega) \qquad (2)$$

$$f_3(\Omega) = \max_{l \in 1,\dots,L} \text{WORKINGMEMORY}_l(\Omega) \qquad (3)$$

where $\text{VALIDATIONACCURACY}(\Omega)$ is the accuracy of the trained model on validation data, $\text{MODELSIZE}(\omega)$, or MS, is the number of bits needed to store the model parameters $\omega$, $\text{WORKINGMEMORY}_l(\Omega)$ is the working memory in bits needed to compute the output of layer $l$, with the maximum taken over the $L$ layers to account for in-place operations. We refer to (3) as the working memory (WM) for $\Omega$. There is no single $\Omega$ which minimizes all of $(1) - (3)$ simultaneously. For instance, (1) prefers large networks with many non-zero weights whereas (2) favors networks with no weights. Likewise, (3) prefers configurations with small intermediate representations, whereas (2) has no preference as to the size of the feature maps. Therefore, in the context of CNN design, it is more appropriate to seek the set of Pareto optimal configurations, where $\Omega^\star$ is Pareto optimal if $f_k(\Omega^\star) \leq f_k(\Omega) \ \forall k, \Omega$ and $\exists j : f_j(\Omega^\star) < f_j(\Omega) \ \forall \Omega \neq \Omega^\star$. The concept of Pareto optimality is appealing for multi-objective optimization, as it allows the ready identification of optimal designs subject to arbitrary constraints in a subset of the objectives.

### 2.1 Search space

Our search space is designed to encompass CNNs of varying depth, width, and connectivity. Each graph consists of optional input downsampling followed by a variable number of blocks, where each block contains a variable number of convolutional layers, each parametrized by its own kernel size, number of output channels, convolution type, and padding. We consider regular, depthwise separable, and downsampled convolutions, where we define a downsampled convolution to be a $1 \times 1$ convolution that downsamples the input in depth, followed by a regular convolution. Each convolution is followed by optional batch-normalization, ReLU, and spatial downsampling through pooling of a variable window size. Each set of two consecutive convolutions has an optional residual connection. Inspired by the decision tree approach in Kumar et al. [36], we let the output layer use features at multiple scales by optionally routing the output of each block to the output layer through a fully connected (FC) layer (see Fig. 1a). All of the FC layer outputs are merged before going through an FC layer that generates the output. The search space also includes parameters governing CNN training and pruning. The Appendix contains a complete description of the search space.

## 2.2 Quantifying memory requirements

The VALIDATIONACCURACY$(\Omega)$ metric is readily available for models via a held-out validation set or by cross-validation. However, the memory constraints of interest in this work demand more careful specification. For simplicity, we estimate the model size as

$$\text{MODELSIZE}(\omega) \approx \|\omega\|_0. \tag{4}$$

For working memory, we consider two different models:

$$\text{WORKINGMEMORY}_l^1(\Omega) \approx \|x_l\|_0 + \|\omega_l\|_0 \tag{5}$$

$$\text{WORKINGMEMORY}_l^2(\Omega) \approx \|x_l\|_0 + \|y_l\|_0 \tag{6}$$

where $x_l$, $y_l$, and $\omega_l$ are the input, output, and weights for layer $l$, respectively. The assumption in (5) is that the inputs to layer $l$ and the weights need to reside in RAM to compute the output, which is consistent with deployment tools like [7] which allow layer outputs to be written to an SD card. The model in (6) is also a standard RAM usage model, adopted in [8], for example. For merge nodes that sum two vector inputs $x_l^1$ and $x_l^2$, we set $x_l = \left[ \left(x_l^1\right)^T \quad \left(x_l^2\right)^T \right]^T$ in (5)-(6). The reliance of (4)-(6) on the $\ell_0$ norm is motivated by our use of pruning to minimize the number of non-zeros in both $\omega$ and $\{x_l\}_{l=1}^L$, which is also the compression mechanism used in related work [36, 24]. Note that (4)-(6) are reductive to varying degrees. However, since SpArSe is a black-box optimizer, the measures in (4)-(6) can be readily updated as MCU deployment toolchains mature.

## 2.3 Neural network pruning

Pruning [37] is essential to MCU deployment using SpArSe, as it heavily reduces the model size and working memory without significantly impacting classification accuracy. Pruning is a procedure for zeroing out network parameters $\omega$ and can be seen as a way to generate a new set of parameters $\bar{\omega}$ that have lower $\|\bar{\omega}\|_0$. We consider both unstructured and structured, or channel [27], pruning, where the difference is that the latter prunes away entire groups of weights corresponding to output feature maps for convolution layers and input neurons for FC layers. Both forms of pruning reduce $\|\omega\|_0$ and, consequently, (4)-(5). Structured pruning is critical for reducing (5)-(6) because it provides a mechanism for reducing the size of layer inputs. We use Sparse Variational Dropout (SpVD) [44] and Bayesian Compression (BC) [42] to realize unstructured and structured pruning, respectively. Both approaches assume a sparsity promoting prior on the weights and approximate the weight posterior by a distribution parameterized by $\phi$. See the Appendix for a description of SpVD and BC. Notably, $\phi$ contains all of the information about the network weight values as well as which weights to prune.

## 2.4 Multi-objective Bayesian optimization

SpArSe consists of three stages, where each stage $m$ samples $T_m$ configurations. At iteration $n$, a new configuration $\Omega^n$ is generated by the multi-objective Bayesian optimizer (MOBO) with probability $\rho_m$ and uniformly at random with probability $1 - \rho_m$. We adopt the combination of model-based and entirely random sampling from [17] to increase search space coverage. The optimizer considers candidates which are morphs of previous configurations and returns both the new and reference configurations (Section 2.5). The parameters of the new architecture are then inherited from the reference before being retrained and pruned.

SpArSe uses a MOBO based on the idea of random scalarizations [47]. The MOBO approach is appealing as it builds flexible nonparametric models of the unknown objectives and enables reasoning about uncertainty in the search for the Pareto frontier. A scalarized objective is given by $g(\Omega) = \max_{k \in 1,...,K} \lambda_k f_k(\Omega)$, where $\lambda_k$ is drawn randomly. Choosing the domain of the prior on $\lambda_k$ allows the user to specify preferences about the region of the Pareto frontier to explore. For example, IoT practitioners may care about models with less than 1000 parameters. Since the functional form of $f_k(\Omega)$ is unknown in practice, it is modeled by a Gaussian process [48] with a kernel $\kappa(\cdot, \cdot)$ that supports the types of variables included in $\Omega$, i.e., real-valued, discrete, categorical, and hierarchically related variables [53, 20]. A new $\Omega^n$ is sampled by minimizing $g(\cdot)$ through Thompson sampling. This MOBO yields better coverage of the Pareto frontier than the deterministic scalarization methods used in [11, 52].

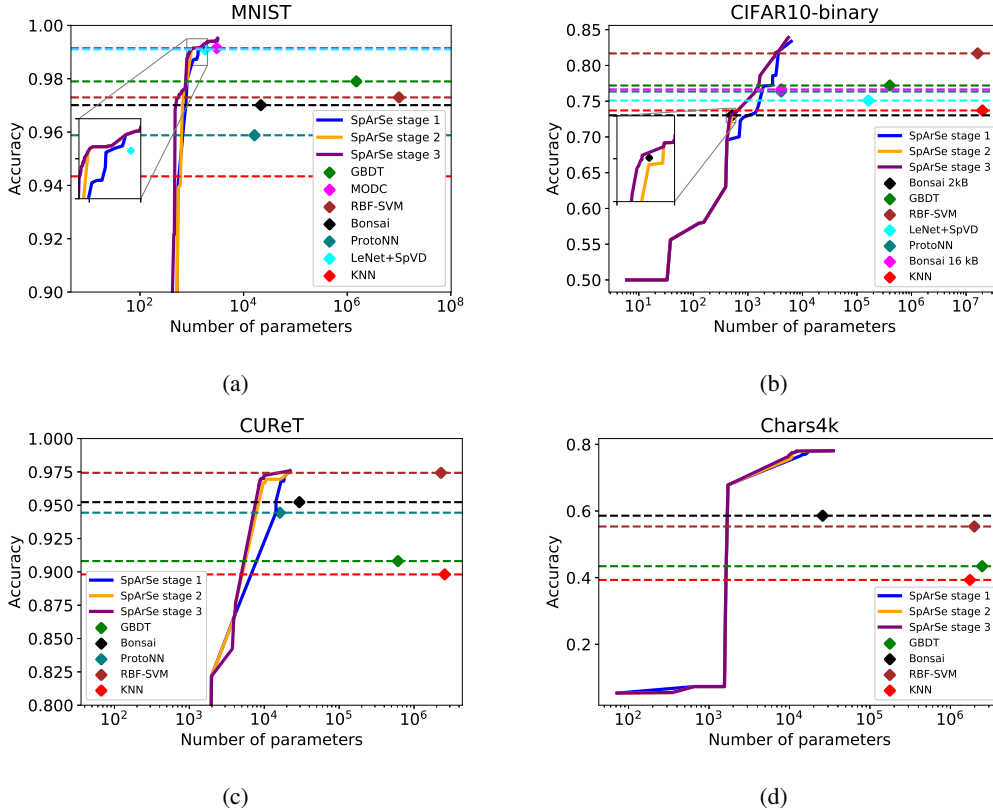

Figure 2: SpArSe results from minimization of $\left(1 - \text{VALIDATIONACCURACY}(\Omega)\right), \text{MODELSIZE}(\omega)$.

## 2.5 Network morphism

Evaluating each configuration $\Omega^n$ from a random initialization is slow, as evidenced by early NAS works which required thousands of GPU days [64, 65]. Search time can be reduced by constraining each proposal to be a morph of a reference $\Omega^r \in \left\{\Omega^j\right\}_{j=0}^{n-1}$ [14]. Loosely speaking, we say that $\Omega^n$ is a morph of $\Omega^r$ if most of the elements in $\Omega^n$ are identical to those in $\Omega^r$. The advantage of using morphism to generate $\Omega^n$ is that most of $\phi^n$ can be inherited from $\phi^r$, where $\phi^r$ denotes the weight posterior parameters for configuration $\Omega^r$. Initializing $\phi^n$ in this way means that $\Omega^n$ inherits knowledge about the value and pruning mask for most of its weights. Compared to running SpVD/BC from scratch, morphisms enable pruning proposals using 2-8× fewer epochs, depending on the dataset. Further details on morphism are given in the Appendix, including allowed morphs.

Because our search space includes such a diversity of parameters, including architectural parameters, pruning hyperparameters, etc., we find it helpful to perform the search in stages, where each successive stage increasingly limits the set of possible proposals. This coarse-to-fine search enables exploring decisions at increasing granularity, to wit: Stage 1) A candidate configuration can be generated by applying modifications to any of $\left\{\Omega^r\right\}_{r=1}^{n-1}$, Stage 2) The allowable morphs are restricted to the pruning parameters, Stage 3) The reference configurations are restricted to the Pareto optimal points.

## 3 Results

We report results on a variety of datasets: MNIST $(55e3, 5e3, 10e3)$ [38], CIFAR10 $(45e3, 5e3, 10e3)$ [34], CUReT $(3704, 500, 1408)$ [57], and Chars4k $(3897, 500, 1886)$ [16], corresponding to classification problems with 10, 10, 61, and 62 classes, respectively, with the training/validation/test set sizes provided in parentheses. To match the setup in [36], we also report on binary versions of these datasets, meaning that the classes are split into two groups and re-labeled. The only pre-processing we perform is mean subtraction and division by the standard deviation. Experiments were run on four

Table 2: Dominating configurations for parameter minimization experiment. SpArSe models are listed on top and the competing method on bottom. SpArSe finds CNNs that are more accurate and have fewer parameters than competing methods. The amount of time spent obtaining each dominating configuration is reported in GPU days (GPUD).

| | MNIST | | | CIFAR10-binary | | | CUReT | | | Chars4k | | |
|---|---|---|---|---|---|---|---|---|---|---|---|---|
| | Acc | $\|\|\bar{\omega}\|\|_0$ | GPUD | Acc | $\|\|\bar{\omega}\|\|_0$ | GPUD | Acc | $\|\|\bar{\omega}\|\|_0$ | GPUD | Acc | $\|\|\bar{\omega}\|\|_0$ | GPUD |
| Bonsai | **97.24** | **510** | 11 | **73.08** | **487** | 1 | **96.45** | **8.5e3** | 1 | **67.82** | **1.7e3** | 1 |
| | 97.01 | 2.15e4 | | 73.02 | 512 | | 95.23 | 2.9e4 | | 58.59 | 2.6e4 | |
| Bonsai (16 kB) | – | – | – | **76.66** | **1.4e3** | 9 | – | – | – | – | – | – |
| | | | | 76.64 | 4.1e3 | | | | | | | |
| ProtoNN | **96.84** | **476** | 11 | **76.56** | **1.4e3** | 10 | **96.45** | **8.5e3** | 1 | – | – | – |
| | 95.88 | 1.6e4 | | 76.35 | 4.1e3 | | 94.44 | 1.6e4 | | | | |
| GBDT | **98.78** | **804** | 11 | **77.90** | **1.6e3** | 8 | **96.45** | **8.5e3** | 1 | **67.82** | **1.7e3** | 1 |
| | 97.90 | 1.5e6 | | 77.19 | 4e5 | | 90.81 | 6.1e5 | | 43.34 | 2.5e6 | |
| kNN | **96.84** | **476** | 11 | **76.34** | **1.4e3** | 10 | **96.45** | **8.5e3** | 2 | **67.82** | **1.7e3** | 1 |
| | 94.34 | 4.71e7 | | 73.70 | 2e7 | | 89.81 | 2.6e6 | | 39.32 | 1.7e6 | |
| RBF-SVM | **97.42** | **569** | 10 | **81.77** | **3.2e3** | 3 | **97.58** | **2.2e4** | 2 | **67.82** | **1.7e3** | 1 |
| | 97.30 | 1e7 | | 81.68 | 1.6e7 | | 97.43 | 2.3e6 | | 48.04 | 2e6 | |
| LeNet + SpVD | **99.16** | **1e3** | 8 | **75.35** | **1.4e3** | 10 | – | – | – | – | – | – |
| | 99.10 | 1.8e3 | | 75.09 | 1.6e5 | | | | | | | |
| MODC | **99.17** | **1.45e3** | 1 | – | – | – | – | – | – | – | – | – |
| | 99.15 | 3e3 | | | | | | | | | | |

NVIDIA RTX 2080 GPUs. We compare against previous SOTA works: Bonsai [36], ProtoNN [24], Gradient Boosted Decision Tree Ensemble Pruning [12], kNN, radial basis function support vector machine (SVM), and MODC [25]. We do not compare against previous NAS works because they have not addressed the memory-constrained classification problem addressed here.

## 3.1 Models optimized for number of parameters

First, we address **C2** by showing that SpArSe finds CNNs with higher accuracy and fewer parameters than previously published methods. We use unstructured pruning and optimize $\{f_k(\Omega)\}_{k=1}^2$. Fig. 2 shows the Pareto curves for SpArSe and confirms that it finds smaller and more accurate models on all datasets. For each competing method, we also report the SpArSe-obtained configuration which attains the same or higher test accuracy and minimum number of parameters, which we term the dominating configuration. Results are shown in Table 2. To confirm that SpArSe learns non-trivial solutions, we compare with applying SpVD pruning to LeNet in Fig. 2 and Table 2.

## 3.2 Models optimized for total memory footprint

Next, we demonstrate that SpArSe resolves **C1**-**C2** by finding CNNs that consume less device memory than Bonsai [36]. We use structured pruning and optimize $\{f_k(\Omega)\}_{k=1}^3$. We quantize weights and activations to one byte to yield realistic memory calculations and for fair comparison with Bonsai [5]. Table 3 compares SpArSe to Bonsai in terms of accuracy, MS, and WM under the model in (5). For all datasets and metrics, SpArSe yields CNNs which outperform Bonsai. For MNIST, Bonsai reports performance on a binarized dataset, whereas we use the original ten-class problem, i.e., we solve a significantly more complex problem with fewer resources. Table 4 reports results for WM model (6), showing that SpArSe outperforms Bonsai across all metrics and datasets, with the exception that Bonsai yields a model with smaller MS for CIFAR10-binary.

## 3.3 What SpArSe reveals about pruning

Pruning can be considered a form of NAS, where $\bar{\omega}$ represents a sub-network of $\{\alpha, \vartheta, \omega\}$ given by $\{\{V, E_p\}, \vartheta, \omega\}$, and $E_p \subseteq E$ contains only the edges for which $\bar{\omega}$ is non-zero [18]. The question then becomes, should one look for $E_p$ directly or begin with a large edge-set $E$ and prune it? There is conflicting evidence whether the same validation accuracy can be achieved by both approaches [18, 19, 41]. Importantly, previous NAS approaches have focused on searching for $E_p$ directly by using $|E|$ as one of the optimization objectives [14]. On the other hand, SpArSe is able to explore both strategies and learn the optimal interaction between network graph $\alpha$, operations $\vartheta$, and pruning. Fig.

Table 3: Comparison of Bonsai with SpArSe for WM model (5). The first row shows the highest accuracy model for WM $\leq$ 2KB and the second row shows the highest accuracy model for WM, MS $\leq$ 2KB. For MNIST, SpArSe is evaluated on the full ten-class dataset whereas Bonsai reports on a reduced two-class problem. SpArSe finds models with smaller MS, less WM, and higher accuracy in all cases. WM,MS reported in KB. Best performance highlighted in bold.

| | MNIST | | | | CIFAR10-binary | | | | CUReT-binary | | | | Chars4K-binary | | | | USPS-binary | | | |
|---|---|---|---|---|---|---|---|---|---|---|---|---|---|---|---|---|---|---|---|---|
| | Acc | WM | MS | GPUD | Acc | WM | MS | GPUD | Acc | WM | MS | GPUD | Acc | WM | MS | GPUD | Acc | WM | MS | GPUD |
| SpArSe | **98.64** | 1.96 | 2.77 | 1 | **73.84** | 1.28 | 0.78 | 5 | **80.68** | 1.66 | 2.34 | 1 | **77.78** | 0.72 | 0.46 | 1 | **96.76** | 1.06 | 1.60 | 1 |
| SpArSe | 96.49 | **1.33** | **1.44** | 1 | **73.84** | 1.28 | 0.78 | 5 | 79.97 | **1.43** | **1.69** | 1 | **77.78** | 0.72 | 0.46 | 1 | **96.76** | 1.06 | 1.60 | 1 |
| Bonsai | 94.38* | < 2 | 1.96 | | 73.02 | < 2 | 1.98 | | – | – | – | | 74.28 | < 2 | 2 | | 94.42 | <2 | 2 | |

Table 4: SpArSe versus Bonsai for WM model (6). See Table 3 for details.

| | MNIST | | | | CIFAR10-binary | | | | CUReT-binary | | | | Chars4K-binary | | | | USPS-binary | | | |
|---|---|---|---|---|---|---|---|---|---|---|---|---|---|---|---|---|---|---|---|---|
| | Acc | WM | MS | GPUD | Acc | WM | MS | GPUD | Acc | WM | MS | GPUD | Acc | WM | MS | GPUD | Acc | WM | MS | GPUD |
| SpArSe | **97.03** | 1.38 | 15 | 1 | **73.66** | 1.13 | 3.95 | 25 | **73.22** | 1.9 | 0.14 | 2 | **76.83** | 0.39 | 20.12 | 1 | **97.56** | 1.81 | 31.79 | 1 |
| SpArSe | 95.76 | **0.62** | **1.76** | 2 | 71.76 | 1.40 | **1.88** | 27 | **73.22** | 1.9 | 0.14 | 2 | 74.87 | 1.64 | **0.16** | 3 | 96.21 | **0.98** | **1.48** | 1 |
| Bonsai | 94.38* | < 2 | 1.96 | | 73.02 | < 2 | 1.98 | | – | – | – | | 74.71 | < 2 | 2 | | 94.42 | <2 | 2 | |

3a compares SpArSe to SpArSe without pruning on MNIST. The results show that including pruning as part of the optimization yields roughly an $80x$ reduction in number of parameters, indicating that the formulation of SpArSe is better suited to designing tiny CNNs compared to [14]. To gain more insight, we show scatter plots of $|E|$ versus $\|\bar{\omega}\|_0$ for the best-performing configurations on two datasets in Fig. 3b-3c, revealing two important trends (see the Appendix for results on the Chars4k and CUReT datasets). First, $\|\bar{\omega}\|_0$ tends to increase with increasing $|E|$ for $|E|$ greater than some threshold $\zeta$. This suggests that optimizing $|E|$ can be a proxy for optimizing $\|\bar{\omega}\|_0$ when targeting large networks. At the same time, $\|\bar{\omega}\|_0$ tends to decrease with increasing $|E|$ for $|E| < \zeta$, which has implications for both NAS and pruning in the context of small CNNs. Fig. 3b-3c suggest that $|E|$ is not always indicative of weight sparsity, such that minimizing $|E|$ would actually lead to ignoring graphs with more edges but the same amount of non-zero weights. Since CNNs with more edges contain more subgraphs, it is possible that one of these subgraphs has better accuracy and the same number of non-zero weights as the subgraphs of a graph with less edges. The key is that pruning provides a mechanism for uncovering such high performing subgraphs [18].

### 3.4 Ablation study

Table 5 presents an ablation experiment on SpArSe with MNIST where we replaced the multi-objective optimizer with a product scalarizer [11, 28] and excluded pruning from the search [13]. In both cases, the algorithm was incapable of finding architectures that are both accurate and meet strict MCU memory requirements. These results support the design choices made in SpArSe in the context of memory constrained MCUs. Table 5 also shows that searching without morphisms yields higher accuracy while meeting the same constraints, albeit at the cost of 50% longer search.

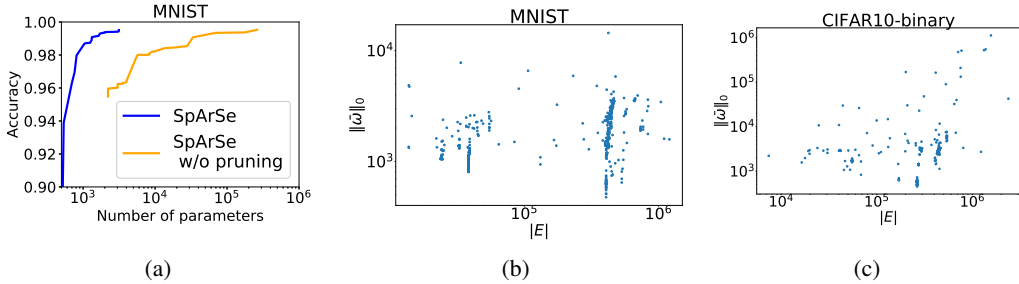

(a)  (b)  (c)

Figure 3: Fig. 3a: Pareto frontier of SpArSe with and without pruning, where both experiments sample the same number (325) of configurations. Fig. 3b-3c show scatter plots of $|E|$ versus $\|\bar{\omega}\|_0$ for the best performing configurations from the parameter minimization experiment. Fig. 3b: MNIST networks with $> 95\%$ accuracy. Fig. 3c: CIFAR10-binary networks with $> 70\%$ accuracy.

Table 5: Ablation on MNIST using WM model (6), searching for models with WM,MS $\leq$ 2KB on 250 configuration budget. SpArSe w/o pruning did not yield a model that satisfies the constraints.

|  | SpArSe | SpArSe w/o pruning | SpArSe w/ product scalarization | SpArSe w/o morphism |
|---|---|---|---|---|
| Acc | 95.76 | – | 11.35 | 97.46 |
| WM | 0.62 | – | 0.01 | 0.68 |
| MS | 1.76 | – | 0.05 | 1.31 |
| GPUD | 2 | – | 2 | 3 |

Table 6: Measurement of SpArSe models on Micro Bit and STM MCUs, compared with Bonsai on Arduino Uno. Latency in ms.

|  | MNIST | | | | | | | CIFAR10-binary | | | | | | | CUReT-binary | | | | | | | Chars4K-binary | | | | | | |
|---|---|---|---|---|---|---|---|---|---|---|---|---|---|---|---|---|---|---|---|---|---|---|---|---|---|---|---|---|
|  | Acc | WM | MS | Lat. $\mu$Bit | ml/inf $\mu$Bit | Lat. STM | ml/inf STM | Acc | WM | MS | Lat. $\mu$Bit | ml/inf $\mu$Bit | Lat. STM | ml/inf STM | Acc | WM | MS | Lat. $\mu$Bit | ml/inf $\mu$Bit | Lat. STM | ml/inf STM | Acc | WM | MS | Lat. $\mu$Bit | ml/inf $\mu$Bit | Lat. STM | ml/inf STM |
| SpArSe | 96.97 | 1.32 | 15.86 | – | – | 285.82 | 203.79 | 73.4 | 2.4 | 9.94 | – | – | 2529.84 | 1803.78 | 73.22 | 2.06 | 0.56 | 671.72 | 70.87 | 103.67 | 73.92 | 74.87 | 1.87 | 0.27 | 207.04 | 21.83 | 77.89 | 55.54 |
| SpArSe | 95.76 | 0.71 | 2.35 | 115.40 | 12.17 | 27.06 | 19.29 | 70.48 | 2.12 | 2.74 | – | – | 498.57 | 355.48 | 73.22 | 2.06 | 0.56 | 671.72 | 70.87 | 103.67 | 73.92 | 74.87 | 1.87 | 0.27 | 207.04 | 21.83 | 77.89 | 55.54 |
| Bonsai | 94.38* | < 2 | 1.96 | 8.9 | 2.18 | 8.9 | 2.18 | 73.02 | < 2 | 1.98 | 8.16 | 2.01 | 8.16 | 2.01 | – | – | – | – | – | – | – | 74.71 | < 2 | 2 | 8.55 | 2.1 | 8.55 | 2.1 |

## 3.5 Latency and power measurements

For validation, we use uTensor [7] to convert CNNs from SpArSe into baremetal C++, which we compile using mbed-cli [3] and deploy on the Micro Bit and STM32F413 MCUs. Table 6 shows the latency and energy per inference measurements. Since uTensor has limited operator support, some networks reported in Table 6 differ from Table 4. Due to uTensor issues with memory management, including memory leaks, some models were only able to be run on the larger MCU. Corresponding measurements for Bonsai cannot be directly compared because Bonsai operates on extracted features instead of the raw input image itself [41]. A recent related work, MODC [25], is considerably slower than SpArSe, at 684 ms for MNIST on the Arduino Uno. Although it may be too early to say if CNN latency/power consumption can meet application requirements, we hope this work provides much needed data to start to answer this question.

## 4 Conclusion

Although MCUs are the most widely deployed computing platform, they have been largely ignored by ML researchers. This paper makes the case for targeting MCUs for deployment of ML, enabling future IoT products and usecases. We demonstrate that, contrary to previous assertions, it is in fact possible to design CNNs for MCUs with as little as 2KB RAM. SpArSe optimizes CNNs for the multiple constraints of MCU hardware platforms, finding models that are both smaller and more accurate than previous SOTA non-CNN models across a range of standard datasets.

### 4.1 Acknowledgements

We thank Michael Bartling, Patrick Hansen, and Neil Tan for their help in model deployment.

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
