[Supplementary Material]

# Appendix A    Pruning algorithm details

Pruning can be expressed as

$$\bar{\omega} = \underset{\left(\sum_{G \in \mathscr{G}} \mathbb{1}\left[\|\omega_G\|_2 > 0\right]\right) \leq s}{\arg\min} \mathscr{L}\left(\{\alpha, \vartheta, \omega\}\right) \tag{7}$$

where $\mathscr{L}(\cdot)$ denotes the loss function for the appropriate task, e.g. cross-entropy for classification, $\mathscr{G}$ denotes the set of disjoint groups covering the indices of each entry in $\omega$, $\omega_G$ denotes a particular group of weights, and $\mathbb{1}[\cdot]$ denotes the indicator function. When $|G| = 1 \forall G \in \mathscr{G}$, (7) is referred to as unstructured pruning. On other other hand, structured pruning arises when $\mathscr{G}$ is chosen to group related elements of $\omega$, i.e. the weights corresponding to a given feature map.

An alternative to (7) is to cast pruning as Bayesian inference with priors that promote sparse solutions [54]. One such algorithm for unstructured pruning is sparse variational dropout (SpVD) [43]. The prior over $\omega$ is assumed to factor over the elements of $\omega$, with $p(|\omega_{ij}|) \propto |\omega_{ij}|^{-1}$. Given a dataset $\mathscr{D}$, the goal of Bayesian inference is to then compute the posterior $p(\omega|\mathscr{D})$. SpVD employs variational inference (VI) [30] to approximate the posterior by a parametrized distribution $q_\phi(\omega)$, whose parameters $\phi$ are chosen to minimize $D_{KL}(q_\phi(\omega)||p(\omega|\mathscr{D}))$. The distribution $q_\phi(\omega)$ is assumed to factor over the elements of $\omega$ and $q_\phi(\omega_{ij}) = \mathsf{N}\left(\mu_{ij}, \beta_{ij}\mu_{ij}^2\right)$, where $\phi = \{\mu, \beta\}$. Techniques for scalable VI are employed to estimate $\phi$ [31, 32]. Upon convergence, the estimate of $\bar{\omega}_{ij}$ becomes $\bar{\omega}_{ij} = \mu_{ij} \odot \mathbb{1}\left[\beta_{ij} \leq \tau_l\right]$, where $\tau_l$ is a layer-specific threshold and $\omega_{ij}$ resides in network layer $l$. Note that $\phi$ contains all of the information about both the network weight values as well as which weights can be masked to $0$. One of the side-effects of the choice of prior in SpVD is that the VI objective decomposes into a sum of a data-dependant term and a term which only depends on the prior, leading to the interpretation of VI as regularized training. Although there is no constant in front of the prior term, it can be beneficial to scale it by $\gamma$. Depending on the dataset, Molchanov et al. [43] keep $\gamma$ at 0 for $N_1$ epochs, which is referred to as the pretraining phase, and then increase $\gamma$ to $\gamma_{N_2}$ over $N_2$ epochs [50]. We include $\{\tau_l\}_{l=1}^{L}$, $N_1$, $N_2$, and $\gamma_{N_2}$ in $\Omega$.

The structured pruning extension of SpVD is called Bayesian Compression (BC) [41], which assumes a hierarchical prior on $\omega$ that ties weights in the same group to each other: $\omega|z \sim \prod_{G \in \mathscr{G}} \prod_{(ij) \in G} \mathsf{N}(\omega_{ij}; 0, z_G^2)$. Inference for this prior proceeds in much the same way as SpVD and, upon convergence, entire groups of weights can be pruned away.

# Appendix B    Search space details

The search space considered in this work is described in Table 7.

# Appendix C    Morphism detals

In the present work, a configuration $\Omega^n$ is considered a morph of $\Omega^r$ if $\Omega^n$ is generated by applying one or more of the operations listed in Table 8 to $\Omega^r$. These morphs are used to generated random samples for the Thompson sampling step in Section 2.4. Each sample $\Omega^n$ is generating by randomly choosing one or more of the morphs from Table 8 and applying them to a randomly chosen $\Omega^r \in \{\Omega^r\}_{r=1}^{n-1}$. This procedure ensures that each configuration proposal is relatively close to a reference configuration. We then use the fact that $\Omega^n$ is closely related to $\Omega^r$ during the pruning process by letting $\phi^n$ inherit information from $\phi^r$, where $\phi^n$ denotes the parameters of the approximated weight posterior for configuration $\Omega^n$. The inheritance process proceeds by first checking for identical nodes between $\Omega^r$ and $\Omega^n$ and then copies the corresponding elements of $\phi^r$ into $\phi^n$ for those nodes. The nodes which participate in this step are the nodes which were not influenced during the morphing process. For the remaining nodes, if corresponding nodes in $\Omega^n$ and $\Omega^r$ have the same operation type, we copy as many of the corresponding elements of $\phi^r$ into $\phi^n$ as possible. For example, if the first layer of $\Omega^n$ is a $3 \times 3 \times 50$ convolution and the first layer of $\Omega^r$ is a $3 \times 3 \times 30$ convolution, we copy the elements of $\phi^r$ corresponding to the first convolution layer into the elements of $\phi^n$ corresponding to the first 30 feature maps of the first convolution layer. Upon completion of the inheritance process, most of the elements of $\phi^n$ are inherited from $\phi^r$, and the remaining elements are learned from the training data. Unlike Elsken et al. [14], we do not restrict the training process to just the elements of $\phi^n$ which were not inherited, but instead update all of the elements of $\phi^n$ during learning.

Table 7: Search space details. For discrete variables, ranges are listed in format [lower-bound:increment:upperbound].

| Name | Range | Description |
|------|-------|-------------|
| downsample-input-in-depth | True/False | If True, max pool the input across the 3rd dimension |
| downsample-input | True/False | If True, max pool the input in spatial dimensions |
| input-downsampling-rate | [2 : 1 : 4] | Active only if downsample-input = True. The amount by which to downsample the input. |
| zero-regularization-epochs | [5:1:30] | Number of epochs for which VI inference is performed before the effect of the sparsity promoting prior is introduced. |
| annealing-epochs | [15:1:25] | Only active if pretraining=False. Number of epochs over which the coefficient in front of the regularization term in the VI objective is annealed from 0 to its final value |
| $\alpha$ | [1e-2:1e-2:1] | Final value for the coefficient of the regularization term in the VI objective |
| pretraining | True/False | Only active if pretraining=False. If True, pretrain the CNN before pruning |
| batch-norm | True/False | Only used for random weight pruning experiments. If True, apply batch-normalization to the output of each layer |
| num-conv-blocks | [1:1:2] | Number of convolution blocks in the CNN, where each block consists of a series of convolutional layers. The output of each block is downsampled through max pooling |
| num-fc-layers | [0:1:1] | Number of FC layers in the main branch following the convolution blocks |
| pruning-thresholds-block-$k$-layer-$l$ | [-6:1e-1:3] | Thresholds for pruning weights in block $k$ layer $l$ |
| total-fc-layer-weights | [1:1:800]e3 | Number of weights in the FC layers comprising the main, left, and right branches |
| weight-fraction-main-branch | [0:1] | Percentage of total-fc-layer-weights that go into the FC layer in the main branch |
| num-conv-layers-block-$k$ | [1:1:3] | Number of convolutional layers in block $k$ |
| layer-type-block-$k$-layer-$l$ | [Conv2D, DownsampledConv2D, SeparableConv2D] | Layer type for convolutional block $k$ layer $l$ |
| kernel-size-block-$k$-layer-$l$ | [2:1:5] | Convolutional kernel size of block $k$ layer $l$ |
| num-filters-block-$k$-layer-$l$ | [1:1:100] | Number of output feature maps for block $k$ layer $l$ |
| downsample-block-$k$-layer-$l$ | [0:0.5] | Active only if layer-type-block-$k$-layer-$l$=DownsampledConv2D. If True, the input feature maps are first passed through a $1 \times 1 \times (downsample - block - k - layer - l \times num - filters - block - k - layer - (l - 1))$ convolutional layer |
| left-branch | True/False | If True, a branch is added to the feed-forward architecture. The branch takes the output of the first convolution block, sends it to an FC layer, sends the result to a merge operation, whose output is sent to a final FC layer |
| right-branch | True/False | If True, a branch is added to the feed-forward architecture. The branch takes the input to the first convolution block, sends it to an FC layer, sends the result to a merge operation, whose output is sent to a final FC layer |
| weight-fraction-left-branch | [0.01:1] | Active only if left-branch=True. Percentage of total-fc-layer-weights that go into the left branch FC layer |
| weight-fraction-left-branch | [0.01:1] | Active only if right-branch=True. Percentage of total-fc-layer-weights that go into the right branch FC layer |
| merge-type | Sum/Concatenate | Active only if at least one of left-branch or right-branch are True. How the main, left, and right branches are to be combined |

Table 8: Allowable morphs. Random sampling is always performed under a uniform distribution.

| Morph | Description |
|---|---|
| num-fc-layers | Change the number of FC layers in the main branch by $\pm 1$ |
| num-conv-blocks | Change the number of convolution blocks by $\pm 1$. If the number of convolution blocks is increased, set the number of convolution layers in the new block to $1$ |
| layer-type | Change the layer type for a randomly chosen convolution layer |
| num-conv-filters | Change the number of output feature maps in a randomly chosen convolution layer |
| kernel-size | Change the kernel size for a randomly chosen convolution filter |
| downsampling-rate | Randomly choose a convolution layer that has type Downsampled-Conv2D and randomly sample its downsampling rate |
| batch-norm | Switch the state of the batch-norm parameter |
| residual-connections | Switch the state of the residual connections parameter |
| left-branch | Switch the state of the left-branch parameter. If the new state is True, set weight-fraction-left-branch=0.05 |
| right-branch | Switch the state of the right-branch parameter. If the new state is True, set weight-fraction-right-branch=0.05 |
| total-fc-layer-weights | Change the number of total FC layer weights by $\pm 5e3$ |
| merge-type | Switch the state of the merge-type parameter |
| threshold | Change the value of a randomly chosen pruning threshold by $0.5$ |
| weight-fraction | For each one weight-fraction-main-branch, weight-fraction-left-branch, weight-fraction-right-branch, perturb each active parameter by 5e-2 |
| $\alpha$ | Change $\alpha$ by $\pm 0.1$ |
| num-conv-layers | Change the number of convolution layers in a randomly chosen convolution block by $\pm 1$ |

(a)                                          (b)

Figure 4: Scatter plots of $|V|$ versus $\|\bar{\omega}\|_0$ for the best performing configurations.

## Appendix D    Visualization of discovered CNNs

Fig. 6 shows the architectures which dominated the competing methods in Table 2.

## Appendix E    Extended results on interaction of pruning and architecture

Fig. 4 shows the interaction of pruning with architecture for the Chars4k and CUReT dataset experiments in Section 3.1. MNIST and CIFAR10-binary are given in Fig. 3.

## Appendix F    Evolution of winning CNNs

The evolution of the CNN architectures which ended up dominating the competing methods on the MNIST dataset in Table 2 is shown in Fig. 5.

Figure 5: MNIST: Evolution of dominating configuration. Lighter colored samples indicate configurations which were sampled later in the optimization process.

(a) Winner against Bonsai on MNIST

(b) Winner against ProtoNN on MNIST

(c) Winner against LeNet+SpVD on MNIST

(d) Winner against GBDT on MNIST

Figure 6: Visualization of winning CNNs on MNIST classification in Table 2. Working memory is reported for the model in (5). The dominating configuration against KNN is the same as that for ProtoNN. The dominating configuration against RBF-SVM is the same as that for Bonsai.

# Appendix G   Visualization of winning CNNs

Fig. 6 shows the architectures which dominating the competing methods on the task of classifying MNIST with the minimum number of parameters (i.e. Section 3.1).