[Reviews · NeurIPS 2019]

Reviewer 1



Originality: It is a novel combination of well-known techniques. It is novel, but not a radically new idea. Quality: The paper is technically sound, the claim are supported by experimental results. One aspect is missing, though: applications with MCUs normally also have throughput constraints. These are frequently seen in NAS papers as a secondary objective beyond accuracy. This is missing here, both as a optimization constraint as well as in the evaluations. Clarity: In general, the paper is clear. Some minor items are missing, though: 1) what is CIFAR10-binary? 2) What microcontroller is used? Significance: In general, the direction is interesting and the paper will provide a new state-of-the-art comparison point. The lack of code and thus reproducibility will have a strongly limiting impact on the significance of this work.

Reviewer 2



This paper combines architecture search and pruning to find better CNN for memory-limited MCUs. The biggest problem of deploying CNN on MCUs is memory constraints. This paper address this problem by using multi-objective optimization. The CNNs found by this paper are more accurate and smaller, and can run on MCUs. Originality: The MCU target is new for architecture search. However, the constrained search problem is not new to architecture search researchers. Existing works have already addressed the problems like constrained number of parameters, constrained FLOPs, or constrained latency. The authors have to cite these works and compare with them. Other methods used in this paper are all well-known techniques. Quality: This submission is technically sound. The proposed methods are thoroughly evaluated. The only missing component is a comparison with existing architecture search works. Clarity: This paper is well-written and well-organized with adequate background information. Significance: The MCU is an interesting target for deploying CNNs. This work definitely advances the state of the art for MCUs. However, I am still doubt about the use cases of CNNs on MCUs. Can the latency and power consumption meet the requirements? Questions: 1. What is the latency and throughput performance of the proposed CNN compared to baselines? These are very important for applications. It will be better if the paper can add some experiments with regard to these metrics. Overall, the MCU target is interesting but the novelty of this paper is limited.

Reviewer 3



• The quality of writing and explanation of the methodology is well done. • The background info about MCUs in the introduction is helpful in building motivation for the problem, however, it may be useful to sacrifice some of these details for more detail in the methodology sections (specifically sections 3.1, 3.4, 3.5). • The design and description of the "multi-objective optimization" framework is also well done. Specifically, the way the authors encode the objectives in (1)-(3), and the desire for a Pareto optimal configuration is reasonable and seems extendable in future works. However, there are some non-trivial topics in this section that could use some more explanation, including the SpVD and BC pruning methods (S3.3), Thompson sampling (S3.4) to get the next configurations, and the coarse-to-fine search (S3.5). • It seems like morphism is primarily included to minimize search time, however how does the framework work without morphism? Do the resulting configs of a morphism-search and non-morphism-search match? Using morphism, is there a strong bias to the initial configuration? • Line 195, how is f_k( omega ) unknown in practice? You define them in (1)-(3) and they seem directly measurable. • Since SpArSe is a "framework" for CNN design, how would this method scale to larger problems/hardware, e.g. CIFAR-10 and/or ImageNet for mobile phones? Or is the future of this framework somewhat limited to MCU designs? • Multi-objective Bayesian optimization is used as the unsupervised learning algorithm. Although the authors do mention that existing approaches to NAS (i.e. reinforcement learning-based) are much less efficient. However, can these other optimization processes (along with genetic algorithms, etc.) be dropped into the SpArSe framework as well? Or is the MOBO an integral component of the proposed framework?

[Author Response · NeurIPS 2019]

We thank the reviewers for their valuable feedback. This rebuttal includes further experiments to address the reviewers'
remarks, and improved experimental results on CIFAR10-binary, finding a model with 76.83% accuracy and WM $\leq$
2KB and a model with 74.87% accuracy and WM,MS $\leq$ 2KB, both of which outperform Bonsai.

**Latency and power measurements (R1,R2)** Table 1 shows the requested latency & energy per inference measurements
on the micro:bit and STM32F413 MCUs (uTensor toolchain). The uTensor toolchain has limited operator support,
so some networks reported in Table 1 differ from Table 4 of the manuscript. Due to uTensor issues with memory
management, including memory leaks, some models were only able to be run on the larger MCU. Corresponding
measurements for Bonsai cannot be directly compared because Bonsai operates on extracted features instead of the
raw input image itself [41]. A recent related work, MODC [Gural and Murmann, 2019], is considerably slower than
SpArSe, at 684 ms for MNIST on the Arduino Uno. It may be too early to say if CNN latency/power consumption can
meet applications requirements, but we hope this work provides much needed data to start to answer this question.

**Experiments on more MCU datasets (R1)** We ran SpArSe on the USPS dataset, yielding models with WM $\leq$ 2KB
(97.56% accuracy), as well as WM,MS $\leq$ 2KB (96.21%), both of which outperform Bonsai at 94.42%.

**Evidence for which components of SpArSe are critical to its performance and discussion of AMC (R3)** We
performed an ablation experiment on SpArSe with MNIST (Table 2), where we replaced the multi-objective optimizer
with product scalarizer (used in [20] and He et al. [2018]) and excluded pruning from the search [23]. In both cases, the
algorithm was incapable of finding architectures that are both accurate and meet strict MCU memory requirements.
These ablation results support the design choices made in SpArSe in the context of memory constrained MCUs.

**Comparisons to existing approaches, including NAS (R2)** To the best of our knowledge, the only existing works
which have proposed models that fit within the MCU-specific constraints WM,MS $\leq$ 2KB are Bonsai and MODC.
None of the prior NAS works have addressed this problem and the lack of publicly available implementations of those
works makes direct comparison challenging. We have compared with Bonsai in the manuscript and to MODC for this
rebuttal. On MNIST, SpArSe achieves accuracy of 99.17% with 1.45e3 parameters, compared to 99.15% accuracy
with 3e3 parameters for MODC. Note that MODC is complimentary to our work, as it proposes novel convolution
implementations, whereas we use uTensor with unoptimized kernels. Manuscript Fig. 3a and Table 2 demonstrate that
SpArSe would not work with the design choices made in previous NAS works, especially [23].

**Impact of morphism (R3)** Table 2 shows that searching without morphisms yields higher accuracy (97.46% vs.
95.76%), while meeting the same constraints of WM,MS $\leq$ 2KB, albeit at the cost of 50% longer search.

**Reproducability (R1)** We are happy to make the implementation publicly available upon acceptance.

**Limited novelty (R2)** We argue that: 1) SpArSe addresses a significant gap in the community, i.e. model design for
constrained MCUs, which form a large portion of deployed hardware, 2) Although the components used by SpArSe
exist in the literature, the combination is unique and non-trivial as confirmed by our ablation experiment and Fig. 3a.

**What is CIFAR10-binary? (R1)** We use the same problem formulation as Bonsai, defined on manuscript line 220.

**Which MCU is used? (R1)** We have used two MCUs to date, micro:bit and STM32F413.

**Validity of claim on line 66 (R1)** Our claim is true for WM $\leq$ 2KB, but we will revise that sentence for clarity.

**Why is $f_k(\omega)$ unknown in practice? (R3)** We mean that the functional form of $f_k(\omega)$ is unknown, although it can
certainly be evaluated after training the network.

**Extension to larger datasets (R3)** We believe our approach can be scaled to larger problems.

Table 1: Measurement of SpArSe models on micro:bit and STM MCUs, compared with Bonsai on Arduino Uno. Latency in ms.

| | MNIST | | | | | | | CIFAR10-binary | | | | | | | CUReT-binary | | | | | | | Chars4K-binary | | | | | | |
|---|---|---|---|---|---|---|---|---|---|---|---|---|---|---|---|---|---|---|---|---|---|---|---|---|---|---|---|---|
| | Acc | WM | MS | Lat. μBit | ml/inf μBit | Lat. STM | ml/inf STM | Acc | WM | MS | Lat. μBit | ml/inf μBit | Lat. STM | ml/inf STM | Acc | WM | MS | Lat. μBit | ml/inf μBit | Lat. STM | ml/inf STM | Acc | WM | MS | Lat. μBit | ml/inf μBit | Lat. STM | ml/inf STM |
| SpArSe | 96.97 | 1.32 | 15.86 | | | 285.82 | 203.79 | 73.4 | 2.4 | 9.94 | – | | 2529.84 | 1803.78 | 73.22 | 2.06 | 0.56 | 671.72 | 70.87 | 103.67 | 73.92 | 74.87 | 1.87 | 0.27 | 207.04 | 21.83 | 77.89 | 55.54 |
| SpArSe | 95.76 | 0.71 | 2.35 | 115.40 | 12.17 | 27.06 | 19.29 | 70.48 | 2.12 | 2.74 | | | 498.57 | 355.48 | 73.22 | 2.06 | 0.56 | 671.72 | 70.87 | 103.67 | 73.92 | 74.87 | 1.87 | 0.27 | 207.04 | 21.83 | 77.89 | 55.54 |
| Bonsai | 94.38* | < 2 | 1.96 | 8.9 | 2.18 | 8.9 | 2.18 | 73.02 | < 2 | 1.98 | 8.16 | 2.01 | 8.16 | 2.01 | | | | | | | | 74.71 | < 2 | 2 | 8.55 | 2.1 | 8.55 | 2.1 |

Table 2: Ablation study on MNIST using WM model (6), searching for models with WM,MS $\leq$ 2KB on 250 configuration budget.

| | SpArSe | SpArSe w/o pruning | SpArSe w/ product scalarization | SpArSe w/o morphism |
|---|---|---|---|---|
| Acc | 95.76 | N/A | 11.35 | 97.46 |
| WM | 0.62 | N/A | 0.01 | 0.68 |
| MS | 1.76 | N/A | 0.05 | 1.31 |
| GPUD | 2 | N/A | 2 | 3 |

Albert Gural and Boris Murmann. Memory-optimal direct convolutions for maximizing classification accuracy in embedded applications. In *ICML*, pages 2515–2524, 2019.

Yihui He, Ji Lin, Zhijian Liu, Hanrui Wang, Li-Jia Li, and Song Han. Amc: Automl for model compression and acceleration on mobile devices. In *ECCV*, pages 784–800, 2018.


[Meta-Review · NeurIPS 2019]

This is a good application of neural architecture search to finding CNN architectures for memory-limited MCUs. Please include the additional results from your feedback in the final version of the paper. Also please make sure to address some of the specific requests in the reviews, such as R1's request to fix up the references.